# Understanding and Simplifying Architecture Search in Spatio-Temporal Graph Neural Networks

**Zhen Xu**                                                      *xuzhen@4paradigm.com*
*4Paradigm*

**Quanming Yao**[*]                                              *qyaoaa@tsinghua.edu.cn*
*Department of Electronic Engineering*
*Tsinghua University*

**Yong Li**                                                      *liyong07@tsinghua.edu.cn*
*Department of Electronic Engineering*
*Tsinghua University*

**Qiang Yang**                                                   *qyang@cse.ust.hk*
*Department of Computer Science and Engineering*
*Hong Kong University of Science and Technology*

**Reviewed on OpenReview:** *https://openreview.net/forum?id=4jEuiMPKSF*

## Abstract

Compiling together spatial and temporal modules via a unified framework, Spatio-Temporal Graph Neural Networks (STGNNs) have been popularly used in the multivariate spatio-temporal forecasting task, e.g. traffic prediction. After the numerous propositions of manually designed architectures, researchers show interest in the Neural Architecture Search (NAS) of STGNNs. Existing methods suffer from two issues: (1) hyperparameters like learning rate, channel size cannot be integrated into the NAS framework, which makes the model evaluation less accurate, potentially misleading the architecture search (2) the current search space, which basically mimics Darts-like methods, is too large for the search algorithm to find a sufficiently good candidate. In this work, we deal with both issues at the same time. We first re-examine the importance and transferability of the training hyperparameters to ensure a fair and fast comparison. Next, we set up a framework that disentangles architecture design into three disjoint angles according to how spatio-temporal representations flow and transform in architectures, which allows us to understand the behavior of architectures from a distributional perspective. This way, we can obtain good guidelines to reduce the STGNN search space and find state-of-the-art architectures by simple random search. As an illustrative example, we combine these principles with random search which already significantly outperforms both state-of-the-art hand-designed models and recently automatically searched ones. [1]

## 1 Introduction

Multivariate forecasting is a crucial task commonly encountered in our daily life. In addition to classic time series forecasting, spatial correlation has been recently leveraged for more accurate prediction, titled Spatio-Temporal Forecasting. A typical and important example is traffic prediction which forecasts future traffic status (e.g., volume and speed) based on history data (usually the same as prediction targets), moving one step further towards smart transportation and intelligent city (Zhang et al., 2011; Ran & Boyce, 2012;

---

[*]Q. Yao is the correspondence author.
[1]Our code is available at `https://github.com/AutoML-Research/SimpleSTG`.

Nagy & Simon, 2018). Using learning models to better exploit spatio-temporal information is the key challenge (Jiang et al., 2021; Li et al., 2021a).

Early works use classic time series forecasting techniques like ARIMA (Zare Moayedi & Masnadi-Shirazi, 2008) and LSTM (Fu et al., 2016) but do not explicitly consider spatial relation. Motivated by the power of Graph Neural Networks (GNNs) (Scarselli et al., 2009; Kipf & Welling, 2017) modeling relational data, two representative works (Yu et al., 2018; Li et al., 2018) firstly combine GNN and temporal modeling techniques and propose Spatio-Temporal Graph Neural Networks (STGNN). STGNNs simultaneously extract spatial correlation by GNN module and temporal correlation by, like Convolutional Neural Network (CNN) (LeCun et al., 1989) or Gated Recurrent Unit (GRU) (Cho et al., 2014), and they have shown promising performance in traffic prediction. Subsequently, many variants are proposed to improve upon above pioneer works from various angles. Examples are STGCN (Yu et al., 2018), DCRNN (Li et al., 2018), MTGNN (Wu et al., 2020), AGCRN (Bai et al., 2020), and STFGCN (Li & Zhu, 2021).

As STGNNs go more complex, designing better architectures by hand can go beyond human expertise because it remains difficult for researchers to understand which architecture configurations work better. More recently, Neural Architecture Search(NAS) methods have gained much attention due to their effectiveness in various research directions (White et al., 2021; Eliasof et al., 2021; Liu et al., 2019). NAS researchers leverage the expertise in certain domains to propose efficient search space and strategy. Similarly, two STGNN NAS methods have been introduced as well AutoCTS (Wu et al., 2021) and AutoSTG (Pan et al., 2021), which basically follow Darts-like differentiable search space (Liu et al., 2019) with a few customized operations to adapt on STGNN models. However, since there is a lack of *systematic architecture understanding* in the STGNN community, the STGNN NAS methods still suffer from a huge search space, hindering their efficiency and effectiveness. Moreover, all the STGNN NAS methods do not consider the impact of *hyperparameters*, leading to probable evaluation bias.

Above works usually propose a novel architecture (manually or by search) and study this single model in an isolated way. Motivated by recent works that study a series of related works in hindsight to produce general principles (Chu et al., 2021; Gorishniy et al., 2021; Bello et al., 2021; You et al., 2020; Zhang et al., 2022), we aim at obtaining useful understandings to bring forth the next generation of STGNNs, which is an important missing piece in the community.

However, systematic understanding of STGNN architectures is not an easy task. First, the evaluation of architectures is expensive as hyperparameter tuning is required. Training hyperparameters in existing works are often tuned with a small grid inherited from prior studies. But different architectures may not prefer the same setting of hyperparameters and a small grid search cannot fully explore the hyperparameters space (You et al., 2020; Bello et al., 2021). Currently, there is no existing methods that can evaluate and compare architectures in a cheap and fair way. Second, the architecture space is huge and complicated. Classically, control variable methods, which analyze improvements with and without certain microscopic architecture designs, are popularly used (Bai et al., 2020; Song et al., 2020). Reusing such methods here could lead to biased interpretations as they fail to explore different architectures from a distributional perspective.

In this work, we take a step back and revisit the STGNNs. Inspired by the representative literature as illustrated in Figure 1, we propose a disentangled framework composed of disjoint factors regarding the architecture designs. This framework includes design choices of temporal/spatial modules, spatio-temporal order and skip connection. Through the lens of this framework, we dig in depth the principles of different architectural choices. We summarize and quantify empirically the influence of each part. Based on our understanding, we could easily find new STGNN models with a simple method adapted from random search.

Our main contributions are as follows:

- We propose a disentangled framework of Spatio-Temporal Graph Neural Networks containing the designs of spatial/temporal modules, the spatio-temporal order and the skip connection.

- To allow fair and efficient evaluation of different model configurations, we study a much larger hyper-parameter setting and quantify the importance of each choice, which helps to reduce significantly the hyperparameter space.

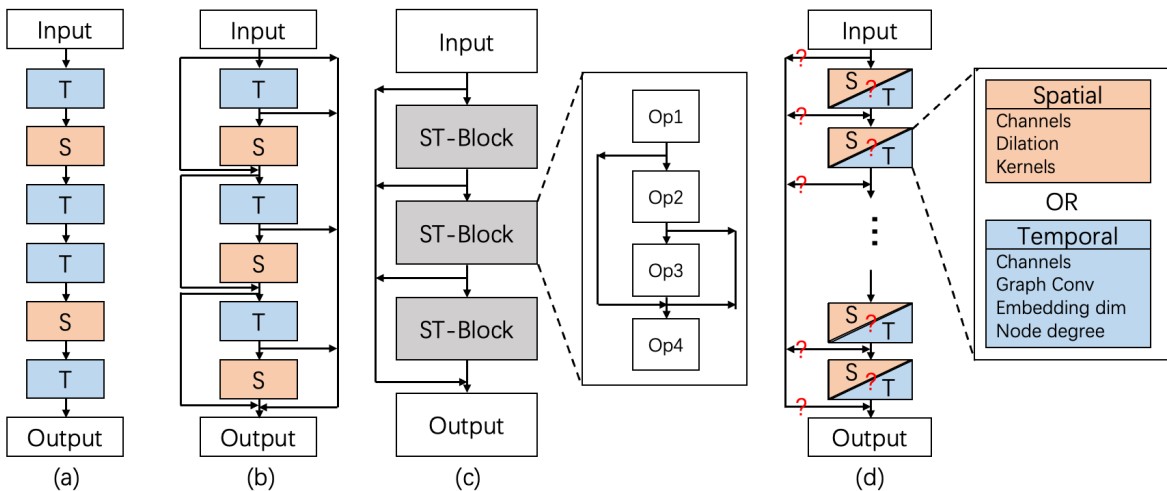

Figure 1: Architecture plot of representative literatures. (a) STGCN (b) MTGNN (c) AutoCTS (d) Our framework. Red blocks represent spatial modules (S). Blue blocks represent temporal modules (T). For AutoCTS, each block composes of four operations and each operation could be spatial or temporal module.

- Through comprehensive experiments and distributional analysis, we conclude fundamental design principles of the skip connection choices and the spatio-temporal order, quantified by mathematically motivated measures. The design principles form naturally cherry regions, where architectures are more likely to perform well.

- Based on our in depth understanding of architectural principles and training hyperparameters, we propose a simple method to obtain new STGNN models more easily and effectively.

## 2   Problem Definition

A graph representing the topology network is denoted as $\mathcal{G} = \{\mathcal{V}, \mathcal{E}, A\}$, where $\mathcal{V}$ is the set of $|\mathcal{V}| = N$ nodes, $\mathcal{E}$ is the edge set and $A$ is the adjacency matrix of shape $\mathbb{R}^{N \times N}$. We have then a multivariate feature tensor of nodes $X \in \mathbb{R}^{T \times N \times D}$ representing the temporal graph signals. $T$ is the total timestamps. $D$ is the number of features per node at each timestamp. Denote also $X_t \in \mathbb{R}^{N \times D}$ the features at one timestamp $t$ and $X_{t:(t+T')} \in \mathbb{R}^{T' \times N \times D}$ the features of $T'$ timestamps.

In spatio-temporal traffic prediction task, the goal is to predict $Q$ future steps based on $P$ past steps observations of traffic features (Li et al., 2021a). Due to the popularity and superior performance, we use STGNNs as the prediction model. Following the mentioned literature (Wu et al., 2020; Chen et al., 2021b; Wu et al., 2021), we consider $P = Q = 12$.

In order to understand the generalizable principles of STGNN architectures in a **tractable** way, we propose the following **distributional** bi-level formulation:

$$\min_\theta \mathbb{E}_{\alpha \sim \mathcal{P}_\theta(\alpha)}[\mathcal{L}(w^*(\alpha), \alpha; D_{\text{val}})] \quad \text{s.t.} \quad w^*(\alpha) = \arg\min_w \mathcal{L}(w, \alpha; D_{\text{train}}),$$

where $\alpha$ is the model architecture, encoded in any raw format; $w$ is the trainable parameters depending on $\alpha$; $\mathcal{P}$ is the architectural distribution parameterized by $\theta$; $\mathcal{L}$ is the evaluation metric such as mean squared error; $D_{\text{val}}, D_{\text{train}}$ are validation and training data. The usage of $D_{\text{val}}$ in the outer optimization and $D_{\text{train}}$ in the inner optimization is commonly adopted in the NAS community in order to seperate conceptually two levels of problem (Liu et al., 2019; Chen et al., 2021a). In our work, we follow this paradigm.

The distributional nature lies in the minimization of expected loss, similar to (Chen et al., 2021a; Xie et al., 2019). And since the raw architecture encoding is not compact and informative enough, we turn this task into

a tractable one by proposing parameterized measures $\theta$ to map from raw architectures $\alpha$ to their measures $\theta$ and finally to the distributional predictive performances.

Generally speaking, the architecture $\alpha$ can capture certain prior knowledge on the traffic prediction task. For example, the order of how spatial/temporal representations are processed, how different levels of spatio-temporal representations could be fused. All these prior knowledge should be data-dependent and can largely influence the architecture choice.

Thus, understanding of architectures leads to better exploit the prior knowledge and subsequently obtain better prediction performance. As discussed in Section 1, existing works cannot be used since the architecture's space is large and complex and a fair evaluation is expensive. In the sequel, we propose a disentangled framework and choose appropriate measures to evaluate architectures on observations generated from the framework.

## 3   A Disentangled Framework

To motivate such a framework, we take a step back and revisit representative works' architectures in Figure 1(a)-(d). We remove unnecessary microscopic details and highlight their overall architectures. STGCN and AGCRN are simplified to compile 6 blocks of Spatial/Temporal modules in different orders (resp. T-S-T-T-S-T and S-T-S-T-S-T). MTGNN compiles 6 blocks in another order (T-S-T-S-T-S) and in addition, adds skip connections. AutoCTS consists of multiple ST-Blocks, each of which consists of 4 operations which could be spatial module, temporal module, identity module, etc. AutoCTS also has fixed skip connections inside the blocks and among the blocks.

After visualizing above literatures, we propose a disentangled framework as shown in Table 1 and Figure 1(e). We are thus interested in the correlation between the performance and these disjoint factors. Specifically, we consider

- *Spatio-temporal order.* Independent of module specific designs, STGNN models put together spatial/temporal modules in a certain order to process spatial and temporal correlations. In our framework, for a pre-determined number of modules, say 6, each module could either be spatial module (S) or temporal one (T). Thus, it is possible to have T-T-T-T-T-T, i.e. all six temporal modules or vice versa. Note that we do not include parallel connections here, e.g., STSGCN (Song et al., 2020) and GMAN (Zheng et al., 2020), because such designs do not perform well, e.g. STFGNN in Table 3.

- *Skip connection.* Earliest works do not consider skip connection (Yu et al., 2018; Li et al., 2018). Later works let temporal modules skip to the end and skip internally every two blocks (Wu et al., 2020). AutoCTS (Wu et al., 2021) connects each block to the end and inside the blocks, every later node connects to all previous nodes in a determined way. We consider a much larger skip connection space of over 2 million choices, i.e., each layer could choose to connect to any number of previous layers.

- *Spatial module.* For graph convolution, we consider several different GCNs that are common in Spatio-temporal GNN literature, namely Kipf GCN, Chebyshev GCN and Mixhop GCN. On graph structure, we adopt the setting of AGCRN but with additional kNN for sparsity. To still make use of the provided graph structure, we include a mixture coefficient.

- *Temporal module.* We only consider TCN here since it is easier to be trained than RNN. Specifically, we consider a typical dilated TCN on the time axis of features to extract temporal features. A set of convolution kernels are used and channels of different kernels are aggregated.

We can see that exemplar hand-designed architectures, e.g., STGCN, MTGNN and searched architectures, e.g., AutoCTS, AutoSTG are all included in the above framework. Moreover, it extends the scope of existing works' to many more possibilities, especially by introducing the space of skip connections and spatio-temporal order.

Table 1: Summarization of the proposed architecture framework

| Type | Name | Detail |
|---|---|---|
| Spatio-temporal order | Arrangement of spatial/temporal modules in arbitrary order | |
| Skip connection | Any jumping between two modules including input and output | |
| Spatial module | Channels | 16,32,64 |
| | Graph convolution | Kipf GCN, Cheb GCN, Mixhop GCN |
| | Embedding dim | 10,20,40,60,80 |
| | Node degree k | 5,10,20,40,60 |
| | Mixture coefficient | 0,0.25,0.5,0.75,1 |
| Temporal module | Channels | 16,32,64 |
| | Dilation | 1,2 |
| | kernel numbers | 1,2,4,8 |
| | kernel candidates | 2,3,4,5,6,7,8,9,10,11,12 |

## 4 Understanding STGNN Hyperparameters and Architectures

In this section, we show how important principles can be observed based on the framework, which help speedup the evaluation and subsequently design better STGNNs. We explain for each factor its motivation, the used methodology and the understanding obtained to answer sequentially the research questions. All experiments in this section are run on two datasets PeMS04 and PeMS08 (Guo et al., 2019; Song et al., 2020), which are introduced with more details in the Appendix B.

### 4.1 Understanding training hyperparameters

Different STGNN models naturally need different training hyperparameters. To avoid biased evaluation of models by inheriting hyperparameter setting as existing works, we instead consider a hyperparameter space where bad hyperparameters choices can be removed such that once we run a model multiple times under this space, we obtain a distributionally fair evaluation.

We consider training hyperparameters (HP) that are common from STGNN literature as in Table 2 where the curriculum learning (Wu et al., 2020; Li et al., 2021a) is specified to STGNN here. The current training hyperparameters constitute a space of over $10^5$ choices and is not practical to efficiently evaluate each model configuration. We aim to independently study the impact by the ranking distribution to remove the hyperparameters that are commonly bad for most architectures. The **ranking distribution** evaluates a model with different hyperparameters to obtain a relative rank and repeats to have distributional patterns. The details of ranking strategy, datasets and choices of hyperparameters are in Appendix B and D.

Table 2: Common training hyperparameters for STGNNs.

| Name | Original Scope | Reduced Scope |
|---|---|---|
| Learning rate | 1e-5, 1e-4, 1e-3, 1e-2, 1e-1 | 1e-4, 1e-3 |
| Batch size | 8,16,32,64,128 | 8, 32, 128 |
| Optimizer | SGD, Rmsprop, Adam, AdamW, Adamax | Adam, AdamW |
| Weight decay | 0,1e-1,1e-2,1e-3,1e-4,1e-5 | 0, 1e-5 |
| Gradient clip | 0,1,3,5,7,9 | 1, 5 |
| Dropout | 0,0.1,0.3,0.5,0.7,0.9 | 0, 0.3 |
| Curriculum learning | None,3,5,7 | None, 3 |

**The ranking strategy.** To compare hyperparameter $P$ of $K$ choices $(p_1, p_2, \ldots, p_K)$, we randomly sample a **configuration** under our framework. A configuration contains all necessary choices of hyperparameters and architectures to identify a model. We then replace iteratively the hyperparameter $P$ of this configuration by all $K$ choices and train the models one by one. Thus, for each batch of $K$ runs, we have $K$ model results

whose configurations differ from each other only in the hyperparameter $P$. We rank these results ascendingly (smaller error metric means better performance, thus ranks better). We run multiple batches of $K$ runs and obtain the ranking distribution on each hyperparameter. For example in Figure 2(a), we compare and rank the relative performance of five learning rate choices for a certain model configuration. After many configurations evaluated, we obtain that 1e-3 is the best choice since it ranks first (most frequent) in all runs.

The ranking plot of each hyperparameter is partially given in Figure 2 and the full plots are in Appendix D. The hyperparameters are grouped into three cases.

1. *reduced options*, e.g., learning rate and optimizer could be reduced to a few options;

2. *monotonically related*, e.g., weight decay and dropout rate are (almost) showing a monotonically better as weight decay decreases;

3. *no obvious pattern*, e.g., gradient clip does not show much difference as long as it is activated and same for batch size and curriculum learning.

The training hyperparameter space has been reduced by 500 times as in Table 2, with which we are able to largely increase the efficiency of model evaluation in a fair way.

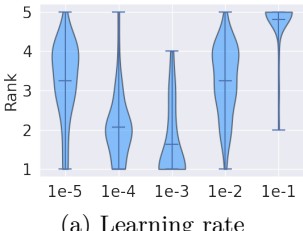
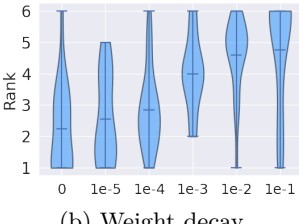
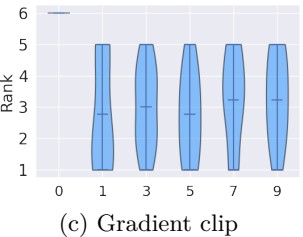

(a) Learning rate    (b) Weight decay    (c) Gradient clip

Figure 2: Rankings of three training hyperparameters. A lower rank means better performance. The more a choice ranks first, the better it is in terms of distribution.

## 4.2 Understanding architectures

With such a disentangled framework presented, we aim at the following research questions (RQ) to understand the influence of each factor:

- **RQ1:** What patterns in terms of spatio-temporal order do good models share? How to characterize spatio-temporal order patterns?

- **RQ2:** What patterns in terms of skip connection do good models share? How to characterize skip connection patterns?

- **RQ3:** How do different spatial/temporal module designs influence the performance?

- **RQ4:** How transferable are above patterns in the architecture space?

However, these questions are not trivial to answer. We need to dedicatedly design systematic views over the huge architecture space such that correlations among different architectures and with the final prediction performance can be simultaneously captured. In this way, principles that can indicate goodness of architectures can be generated. Ideally, we want to find cherry regions of architecture space since good architectures should share important designs in common. An architecture in the cherry region has a high probability to achieve good performance. A key challenge in finding such regions lies in the choice of quantitative measures.

### 4.2.1 Spatio-temporal order

The spatio-temporal order specifies the arrangement of both spatial/temporal modules to learn the representation. Intuitively, the representation is influenced by how many spatial/temporal modules we use respectively

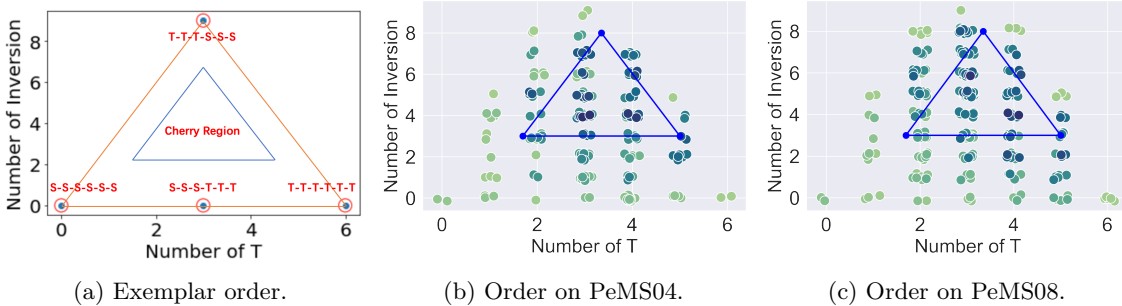

(a) Exemplar order.     (b) Order on PeMS04.     (c) Order on PeMS08.

Figure 3: Spatio-temporal order understanding on different datasets. For (b)(c), we normalize performances to show them in a colormap. The darker the circle, the better the performance.

and how these modules are interleaved to exchange the information. We propose two measures explained below to characterize spatio-temporal order of an architecture:

- **Number of spatial or temporal modules.** Without loss of generality, we observe temporal modules, noted as #T

- **Number of inversed order of a module sequence.** Let the number of inversed order of a pair of modules (either S or T) be 1 if S comes before T, i.e. ST, and 0 otherwise, i.e. TS, SS, TT. The number of inversed order of a module sequence is defined as the sum of that of each pair modules.

The rationale of the above two measures is quantitatively justified in below Remark 1. Specifically, they help us identify a compact triangle region as in Figure 3(a) where different combinatorial choices of S/T orders lie at the corners.

**Remark 1.** *We claim that the number of inversed order along with the number of T modules (#T), can well capture the interleaving of a sequence of S-T modules, including the following 4 corner cases: (N is number of total modules)*

- *Case 1: All S modules, #T is 0 and number of inversed order is 0, indicating no temporal feature is needed;*

- *Case 2: All T modules, #T is N and number of inversed order is 0, indicating no spatial feature is needed;*

- *Case 3: $\frac{N}{2}$ S modules followed by $\frac{N}{2}$ T modules, #T is $\frac{N}{2}$ and number of inversed order is 0, indicating the spatial information should strictly be processed before temporal information;*

- *Case 4: $\frac{N}{2}$ T modules followed by $\frac{N}{2}$ S modules, #T is $\frac{N}{2}$ and number of inversed order is $\frac{N^2}{4}$, indicating the temporal information should strictly be processed before spatial information.*

We fix the framework factors except for the spatio-temporal order. As mentioned above, we relax the spatio-temporal order under 6 temporal and/or spatial modules, leading to a space of $2^6 = 64$ choices. We use the above mentioned measures to observe the order. Results are in Figure 3. We can see that for spatio-temporal order, all corner cases perform terribly. A clear region for 2 to 5 T blocks and for number of inversions between 3 and 8 exists and shows superior performance. This suggests that (1) we might have good performance for many temporal modules but too many spatial modules is not a good idea. This could be because of the oversmoothing phenomenon observed in GNN (Xu et al., 2018) but further study is required; (2) a large number of inversion usually returns better performance which means that both modules are encouraged to interleave more frequently.

### 4.2.2 Skip connection

As the representation flows from the input to the output without a loop, we are motivated to formulate skip connections in STGNNs as a Directed Acyclic Graph (DAG) with a straight though flow path connecting sequentially each module (Figure 1(e)). Afterwards, we need to consider proper measures to observe the skip space as it is large. Intuitively, the performance of a certain skip pattern depends on how many we skip and

where we skip. The former can be measured by number of paths from input module to output module and the latter can be measured by average shortest length for each pair of modules, both defined below. We propose another two measures explained below to characterize skip connection of an architecture:

- **Number of path**, noted as (#Path). It is defined as the total number of trajectories from the predetermined input node to the predetermined output node.

- **Average Shortest Length**, noted as ASL. It is defined as the average of shortest lengths from non-output nodes to the output node predetermined.

The rationale of the above two measures is quantitatively justified in below Remark 2. Specifically, they help us identify a compact right triangle region as in Figure 4(a) where different combinatorial choices of skip connections lie at the corners.

**Remark 2.** *We claim that Average Shortest Length (ASL) and number of path(#Path) in a Directed Acyclic Graph can well capture the spatio-temporal information flow, including the following 3 corner cases:*

- *Case 1: ASL is low and #Path is low, i.e., limited number of skips which mostly connects to the output node, where most literature falls in;*
- *Case 2: ASL is low and #Path is high, i.e., a lot of skip connections among which many skips to the output node, potentially obfuscating the propagated message;*
- *Case 3: ASL is high and #Path is low, i.e., almost no additional skips, which is the case of earliest STGCN model (Yu et al., 2018).*

In Figure 4, we show the skip experiment on two datasets and on two spatio-temporal orders that have been used in the literature. Details are given in Appendix D. All these four experiments follow a similar pattern. First, all corner cases perform badly. In the center rectangle, we observe a clear region where the performance are much better, especially for ASL between 1.1 to 1.8 and #Path between 10 and 40. Notably, this region exists for different datasets and for different spatio-temporal orders, showing its universality in terms of principles. This observation can be exemplified by other works e.g. residual connection (He et al., 2016), JK-Net (Xu et al., 2018), message passing (Battaglia et al., 2018). Especially for GNN, we want to reinforce the remote message, e.g. the input node, in later nodes, thus ASL will be low. We also do not want to have too many messages that obfuscate the valuable message, thus #Path cannot be too high.

### 4.2.3 Spatial/temporal module design

Different choices of spatial/temporal module influence the way the spatial/temporal representation are processed. In the literature, diverse modifications on TCN and GCN have been proposed. We use the same ranking distribution as in understanding training hyperparameters to study all the module choices. The ranking plots are partially shown in Figure 5 and full plot is given in Appendix D. We remove a few bad options according to the ranking, e.g., Cheb GCN, node degree 5, etc. Then, we also find we could largely reduce number of parameters by reducing channels temporarily due to a high performance correlation, as show in Appendix D.

### 4.2.4 Summary of designing principles

With the empirical results in this section, after refining the training hyperparameter space, we find that good architectures share similar patterns. Firstly, Two cherry regions exist as shown in Figure 3(a) and Figure 4(a), answering **RQ1** and **RQ2**. Concretely, we tend to consider architectures which have appropriate number of temporal modules, inversion number, averaged shortest length, and finally number of paths. The appropriate scope can be interpreted from the view of spatio-temporal information flow and message passing with jumping information. For module designs, we remove some bad choices as by ranking and discover that we could largely reduce number of parameters by reducing channel size due to high performance correlation, answering **RQ3**. All the experiments in this section are conducted on two datasets PeMS04 and PeMS08, showing the generability of the discoveries, thus answering **RQ4**.

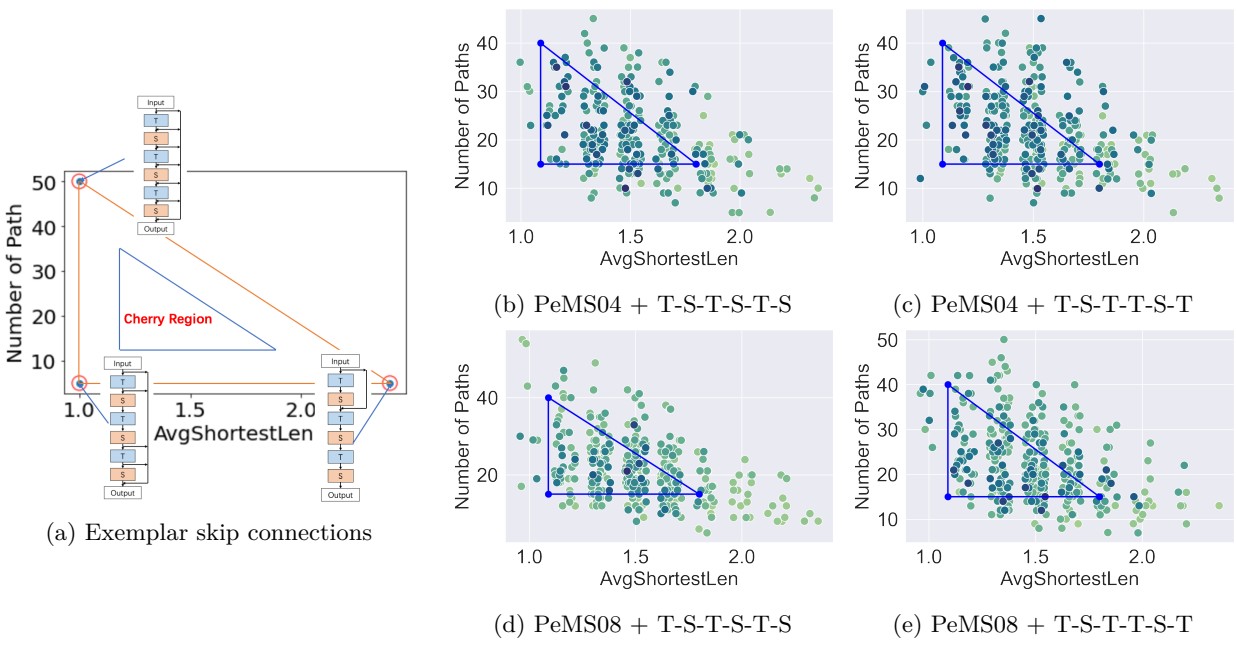

Figure 4: Skip understanding on different spatio-temporal orders and different datasets. For (b)(c)(d)(e), we normalize performances to show them in a colormap. The darker the circle, the better the performance.

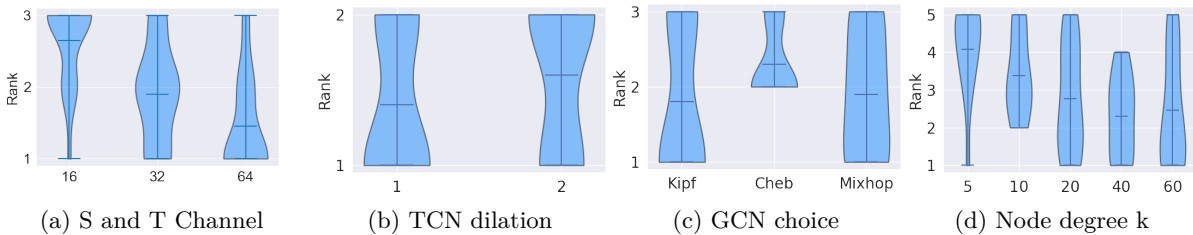

Figure 5: Rankings of several design choices. A lower rank means better performance.

## 5 Searching Better STGNN Models in a Simplified Way

### 5.1 A simplified but strong NAS baseline

Here, to give an example on how to leverage these understandings, we show a simple but strong NAS baseline based on Random Search (RS) that is able to find better STGNN models efficiently, titled **SimpleSTG** as illustrated in Figure 6. We use random search with the cherry regions found by understanding results on typical datasets. Specifically, we evaluate a *random* sample if it lies in regions, otherwise reject it and re-sample. Note that more complex search strategies leveraging our understanding surely exists and probably will be more efficient than our simple baseline, e.g. Evolutionary, Bayesian or even differentiable. But our goal is to demonstrate how to utilize our understanding and how effective it could be to design architectures with better understandings. We stick to this baseline for further comparison.

### 5.2 Overall performance comparison

In this part, we compare more thoroughly our models with related works. From Table 3, incoporating architecture priors, the proposed simple baseline could find novel STGNN models better than hand-designed and NAS-based methods. The other baselines are introduced as follows. Vector Auto-Regression (VAR) is a baseline often for sanity check.

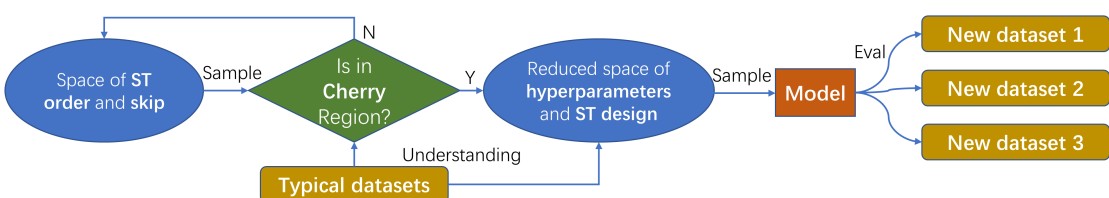

Figure 6: Illustration of the pipeline in our simple random search baseline SimpleSTG.

ASTGCN (Guo et al., 2019) adopts attention mechanism for both spatial and temporal modeling. STS-GCN (Song et al., 2020) proposes synchronous modules to leverage the heterogeneities in spatial-temporal data. STSGCN (Song et al., 2020) stacks multiple localized GCN for extraction of correlations. STGODE (Fang et al., 2021) uses a tensor-based ordinary differential equation. Z-GCNETs (Chen et al., 2021b) is GRU-based and introduces a time zigzag persistence to integrate with GCN. Others have been discussed in Section 1. Note that there are many NAS methods for pure GNN such as (Li et al., 2021b; Zhao et al., 2021). These methods cannot be used directly in STGNN because of the difference in search space.

Table 3: Prediction performance comparison of different models. Bold number denotes the best and underscored number denotes the second best. Numbers in the parentheses indicate the standard deviation.

| Model | PeMS03 | | PeMS04 | | PeMS07 | | PeMS08 | |
|---|---|---|---|---|---|---|---|---|
| | MAE | RMSE | MAE | RMSE | MAE | RMSE | MAE | RMSE |
| VAR | $23.65_{(0.00)}$ | $38.26_{(0.00)}$ | $23.75_{(0.00)}$ | $36.66_{(0.00)}$ | $75.63_{(0.00)}$ | $115.2_{(0.00)}$ | $23.46_{(0.00)}$ | $36.33_{(0.00)}$ |
| STGCN | $17.49_{(0.46)}$ | $30.12_{(0.70)}$ | $22.70_{(0.64)}$ | $35.55_{(0.75)}$ | $25.38_{(0.49)}$ | $38.78_{(0.58)}$ | $18.02_{(0.14)}$ | $27.83_{(0.20)}$ |
| DCRNN | $18.18_{(0.15)}$ | $30.31_{(0.25)}$ | $24.70_{(0.22)}$ | $38.12_{(0.26)}$ | $25.30_{(0.52)}$ | $38.58_{(0.70)}$ | $17.86_{(0.03)}$ | $27.83_{(0.05)}$ |
| ASTGCN | $17.69_{(1.43)}$ | $29.66_{(1.68)}$ | $22.93_{(1.29)}$ | $35.22_{(1.90)}$ | $28.05_{(2.34)}$ | $42.57_{(3.31)}$ | $18.61_{(0.40)}$ | $28.16_{(0.48)}$ |
| STSGCN | $17.48_{(0.15)}$ | $29.21_{(0.56)}$ | $21.19_{(0.10)}$ | $33.65_{(0.20)}$ | $24.26_{(0.14)}$ | $39.03_{(0.27)}$ | $17.13_{(0.09)}$ | $26.80_{(0.18)}$ |
| AGCRN | $16.58_{(0.10)}$ | $27.48_{(0.14)}$ | $19.83_{(0.06)}$ | $32.26_{(0.12)}$ | $24.21_{(0.21)}$ | $37.66_{(0.24)}$ | $15.95_{(0.18)}$ | $25.22_{(0.22)}$ |
| MTGNN | $15.23_{(0.03)}$ | $26.12_{(0.20)}$ | $19.25_{(0.03)}$ | $31.65_{(0.21)}$ | $21.28_{(0.11)}$ | $34.31_{(0.16)}$ | $15.86_{(0.10)}$ | $24.93_{(0.19)}$ |
| STGODE | - | - | $20.84_{(0.00)}$ | $32.82_{(0.00)}$ | - | - | $16.81_{(0.00)}$ | $25.97_{(0.00)}$ |
| Z-GCNETs | - | - | $19.90_{(0.00)}$ | $32.66_{(0.00)}$ | - | - | $16.12_{(0.00)}$ | $25.74_{(0.00)}$ |
| STFGNN | $16.77_{(0.09)}$ | $28.34_{(0.46)}$ | $19.83_{(0.06)}$ | $31.88_{(0.14)}$ | $22.07_{(0.11)}$ | $35.80_{(0.18)}$ | $16.64_{(0.09)}$ | $26.22_{(0.15)}$ |
| AutoCTS | $\underline{14.71}_{(0.40)}$ | $\underline{24.54}_{(0.33)}$ | $\underline{19.13}_{(0.21)}$ | $\mathbf{30.44}_{(0.24)}$ | $\underline{20.93}_{(0.35)}$ | $\underline{33.69}_{(0.29)}$ | $\underline{14.82}_{(0.17)}$ | $\mathbf{23.64}_{(0.10)}$ |
| **SimpleSTG (ours)** | $\mathbf{14.45}_{(0.10)}$ | $\mathbf{24.35}_{(0.13)}$ | $\mathbf{18.56}_{(0.26)}$ | $\underline{30.71}_{(0.41)}$ | $\mathbf{19.80}_{(0.20)}$ | $\mathbf{33.03}_{(0.18)}$ | $\mathbf{14.64}_{(0.10)}$ | $\underline{23.77}_{(0.12)}$ |

The **generalizability** of the distilled principles is demonstrated in two ways. First, in Section 4, we show at the same time empirical results on two datasets and similar principles are observed. Second, in Table 3, note that we search only one model and evaluate this model on more datasets covering different regions of California (Appx B) instead of searching the best model per dataset as in other NAS methods. We show further the comparison on a new and different dataset NE-BJ released by (Li et al., 2021a). The NE-BJ dataset contains traffic information in Beijing, which is totally different from commonly used California datasets. The metric is MAE and RMSE on 15 mins, 30 mins and 60 mins prediction to comprehensively evaluate the effectiveness our our model. In almost all test cases, our baseline outperforms by a large margin. We also provide evaluation on another non-traffic dataset and the configuration of searched model in Appx E.

### 5.3 Search efficiency and effectiveness of space

We further demonstrate the effectiveness of our understanding on training hyperparameters and architectures in Figure 7(a-e) on two datasets. In the Figure 7(a), we evaluate the search efficiency in terms of GPU hours and compare NAS methods with hand designed results on dataset PeMS08. The orange line is our accelerated search in small channel and the green line is the same architecture as in orange line but with large channel size. The motivation and correlation of performances with different channel sizes are illustrated in Appx D. It can be found that our NAS method, though implemented with a simple random search strategy, can be very

Table 4: Performance comparison on NE-BJ. Bold number denotes the best and underscored number denotes the second best. Instead of avearged 12 steps, we show results on 3/6/12 steps each for thoroughness.

| Model | 15 mins | | 30 mins | | 60 mins | |
|---|---|---|---|---|---|---|
| | MAE | RMSE | MAE | RMSE | MAE | RMSE |
| VAR | 5.42 | 8.16 | 5.76 | 9.07 | 6.14 | 9.65 |
| DCRNN | 3.84 | 6.84 | 4.51 | 8.49 | 5.15 | 9.77 |
| STGCN | 5.02 | 8.34 | 5.10 | 8.55 | 5.39 | **9.09** |
| ASTGCN | 4.43 | 7.34 | 5.31 | 8.86 | 6.29 | 10.31 |
| AGCRN | 3.90 | 6.81 | 4.55 | 8.32 | 5.06 | 9.54 |
| GMAN | 4.08 | 7.63 | 4.42 | 8.45 | **4.80** | 9.18 |
| AutoCTS | 3.91 | 6.70 | 4.69 | **8.21** | 5.64 | 9.80 |
| **SimpleSTG (ours)** | **3.71** | **6.69** | **4.33** | **8.21** | 4.85 | 9.34 |

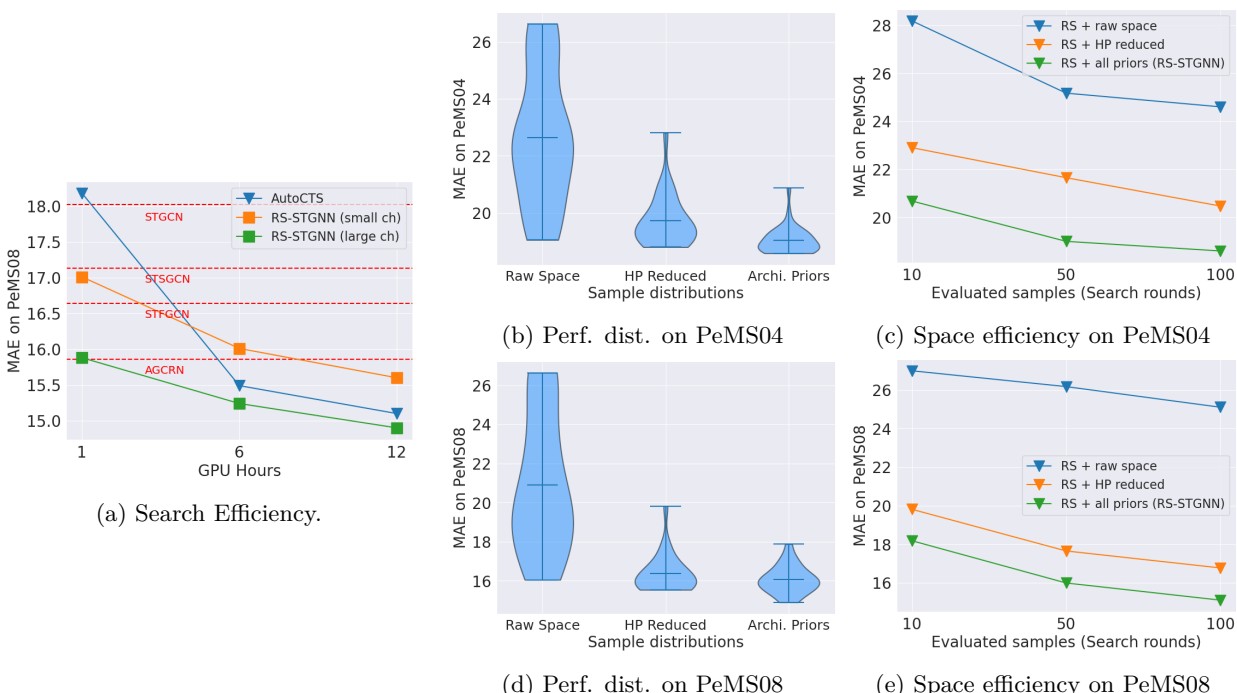

(a) Search Efficiency.

(b) Perf. dist. on PeMS04

(c) Space efficiency on PeMS04

(d) Perf. dist. on PeMS08

(e) Space efficiency on PeMS08

Figure 7: Empirical results on the search efficiency and effectiveness of space.

efficient in terms of GPU hours. Note also that our random search baseline is straightforward to adapt on distributed systems without communication head while AutoCTS cannot easily achieve significant acceleration. In Figure 7(b)(d), we show the samples' performances in different spaces. With architecture priors and reduced hyperparameter space, the performance distribution is significantly better. In Figure 7(c)(e), we show the random search efficiency considering different priors. Both plots show the effectiveness of our understanding and concluded principles.

## 6 Conclusion and discussion

In this work, we revisit Spatio-Temporal Graph Neural Networks in traffic prediction to study how to properly design models in a principled way. We propose a framework that disentangles the choices into three groups: spatial/temporal module designs, spatio-temporal order and skip connection. We understand qualitatively and quantitatively the influence of each disjoint factor and conclude the principles behind the architecture

design. To illustrate how these understandings could help find better STGNN models, we propose a simple strategy to efficiently and effectively search models that outperform all other works.

This work of course has limitations and could be further explored. Firstly, we limit our study scope to traffic prediction while STGNN is also used in other tasks and modalities, e.g. video action recognition, weather forecasting, etc. The STGNNs considered in other communities are not the same thus need further exploration. Besides STGNN, other neural networks might also benefit from similar cherry regions. On the other hand, we consider TCN for temporal modeling in this work due to its efficiency and flexibility in architecture design. It is also interesting to consider RNN-based or attention-based STGNN framework for a more unified understanding. Other future works include the introduction of differentiable search policy with the findings and the consideration of possible confounding factor or dependency of factors.

**Acknowledgment**

Q. Yao was in part supported by NSFC (No. 92270106) and CCF-Baidu Open Fund. This work is also partially supported by the National Key Research and Development Program of China under Grant 2018AAA0101100 and Hong Kong RGC TRS T41-603/20-R.

**Broader Impact Statement**

Our work does not contain potential harms to people, environment, society or any possible negative impacts listed on the TMLR Ethics Guidelines.

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

# A  Related works

Our proposed framework differs from existing works as follows. DL-Traff (Jiang et al., 2021) surveys the literature and proposes a framework based on the modeling method on spatial axis and temporal axis (e.g., CNN, GRU, GNN, etc.). DGCRN (Li et al., 2021a) considers additional spatial topology construction and external features to classify different methods. STMeta (Wang et al., 2021) proposes another analytical framework concerning spatio-temporal information modeling. Our framework differs from then in that we also consider macroscopic factors and training hyperparameters. The purpose of our framework is also different because we use our framework to understand the architecture space which potentially facilitates further architecture search. On the other hand, AutoCTS (Wu et al., 2021) and AutoSTG (Pan et al., 2021) proposes their own framework of STGNN to search architectures, but they do not consider skip connection or training hyperparameters. AutoCTS and AutoSTG do not try to understand the space.

Our methodologies of understanding have been inspired by many other works, including architecture understanding in Convolution Neural Networks, Graph Neural Networks and Neural Architecture Search (You et al., 2020), different communities' common discovery that training hyperparameters once well tuned can boost the model performance by a large margin and probably influence the models' evaluation (Tan & Le, 2019; You et al., 2020). Previous works on surveys, benchmarks, and NAS of traffic prediction should also be credited, as discussed in the previous paragraph.

# B  Details on the datasets and problem setting

## B.1  Dataset

To evaluate quantitatively the performance of different models, we experiment on four public real world datasets: PeMS03, PeMS04, PeMS07 and PeMS08 (Guo et al., 2019; Song et al., 2020; Bai et al., 2020; Fang et al., 2021). These datasets can be accessed on GitHub[2]. These datasets come from Caltrans Performance Measurement System (PeMS) and contain the measures of highway traffic flow in different regions. The raw temporal signal was recorded every 30 seconds and we aggregate both datasets into 5-mins regular traffic measures the same way as previous works. Detailed statistics are given in Table 5. Datasets are splitted in a 6,2,2 manner. A standard z-score normalization is applied which subtracts and divides data by mean and standard deviation of training split. We DO NOT include additional hand-craft features e.g. calendar, holiday or time of the day. In all our experiments of Section 4, we use both PeMS04 and PeMS08 for understanding general principles. In Section 5, we show overall model performance comparisons on all these four datasets.

Table 5: Dataset statistics

| Dataset | Region | #Timestamps | #Sensors | Time Period |
|---------|--------|-------------|----------|-------------|
| PeMS03 | Marysville | 26208 | 358 | 01/09/2018 - 30/11/2018 |
| PeMS04 | Oakland | 16992 | 307 | 01/01/2018 - 28/02/2018 |
| PeMS07 | Los Angeles | 28224 | 883 | 01/05/2017 - 31/08/2017 |
| PeMS08 | San Bernardino | 17856 | 170 | 01/07/2016 - 31/08/2016 |

## B.2  Problem setting

In Section 2, we give a formal problem definition on spatio-temporal traffic prediction. In practice, following the literatue (Yu et al., 2017; Cui et al., 2020; Bai et al., 2020; Wu et al., 2020), we set $P = Q = 12$, i.e., predicting the next one hour based on past hour. Concretely, if we predict on dataset PeMS04 from Table 5, the input tensor is of shape: $(307, 12, 1)$ and the output tensir is of shape: $(307, 12, 1)$. 1 is the channels of traffic signal or feature. Since we do not include additional calendar or hand-designed features, this channel remains 1.

---

[2]https://github.com/Davidham3/ASTGCN/tree/master/data

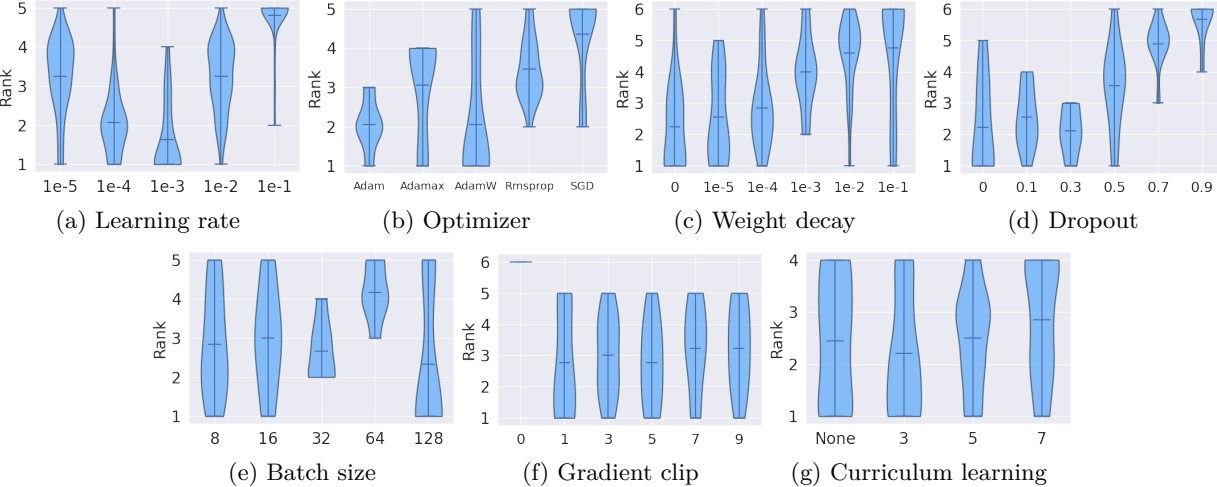

Figure 8: Rankings of different training hyperparameters. A lower rank means better performance. The more a choice ranks first, the better it is in terms of distribution.

## C    Details on the disentangled framework

### C.1    Curriculum learning

As pointed out by (Wu et al., 2020; Li et al., 2021a), curriculum learning improves consistently the multi-step prediction. To use curriculum learning in traffic prediction, we first start with one step prediction with all training data (easiest task) and gradually increase the prediction horizon to deal with more difficult task. This curriculum learning strategy brings another hyperparameter: number of epochs to increase the task difficulty.

### C.2    Calculation of skip connection choices

We have in total $2^{\frac{(L-1)(L-2)}{2}}$ skip connection choices where $L$ is number of modules (including input module, spatio-temporal modules, and output module). The first module has $(L-2)$ skip placeholders. The second module has $(L-3)$ skip placeholders, and so on. In the case of 8 modules (6 spatio-temporal module plus 1 input and 1 output module), we will have a space of $2^{6+5+4+3+2+1} = 2^{21} = 2.09$ millions choices.

## D    Details on the understanding results

### D.1    More observations on training hyperparameters

Additional interesting observations could be made in Figure 2. For example, batch size 64 which most literature uses is worst in our ranking The optimizer AdamW (Loshchilov & Hutter, 2019) is not considered by any literature in traffic prediction but showed best in our rank. All these give evidences to certain biases in training hyperparameters. As a result, to compare fairly different model architectures, we need to take into account the training hyperparameters.

### D.2    More on spatial/temporal module design

Different choices of spatial/temporal module influence the way the spatial/temporal representation are processed. In the literature, diverse modifications on TCN and GCN have been proposed. With limited ablation study, it is hard to conclude which part really contributes to the performance gain, thus hinder our understanding on spatio-temporal modeling.

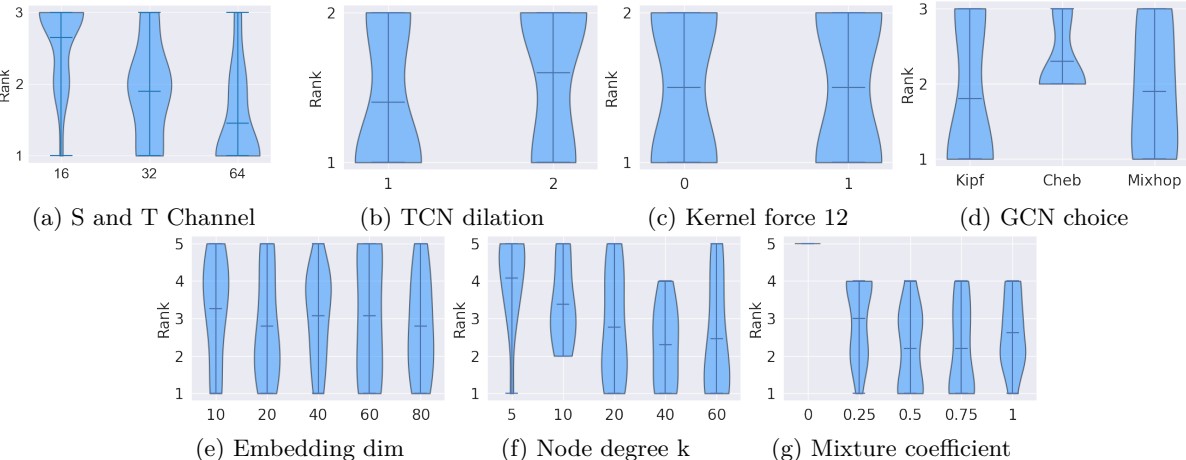

Figure 9: Rankings of different design choices. A lower rank means better performance.

We use the same ranking distribution as in understanding training hyperparameters to study all the module choices. Due to the huge impact on model parameters, we study in addition the channel correlation whose detailed plots are in Appendix D.

The full ranking plots are shown in Figure 9. Firstly, except for spatial/temporal channels, most designs from the literature are useful. We remove a few bad options according to the ranking, e.g., Cheb GCN, zero mixture coefficient, etc. Then, spatial/temporal channel influences a lot the trainable parameters and shows consistent goodness as in Figure 9(a), we study further the performance correlation between channel 64 and channel 32. We find that channel 32 reaches a correlation up to 0.85 with channel 64 while having only half of the parameters. Thus, we use channel 32 for all the intermediate study for acceleration and change back to 64 for final evaluation.

### D.3 More on channel correlation

We have found in Section 4 that channel rank is very significant, i.e., larger channel gives better performance. Actually, it is usually believed that larger channels can perform better but also bring more trainable parameters (Tan & Le, 2019). Thus, after ranking them, we consider in addition to study the performance correlation between different channel choices to understand its impact and further opens the possibility to scale down models to accelerate the execution without messing up models' relative performance. We choose spatial module channel to be the same as temporal module channel for simplicity.

First, the channel correlation can be visualized in Figure 10. We show that on both datasets (PeMS04 and PeMS08), channel 32 achieves a high performance correlation with channel 64 while the correlation between channel 16 and channel 64 are not stable. On the other hand, channel 32 already reduces half of the trainable parameters compared to channel 64 and significantly lowers the training time cost. As a result, we will use channel 32 for the understanding study except for the final performance comparison where we use channel 64.

### D.4 More on the chosen measures

Note that other measures exist but may not be the most appropriate in our study. In the example of skip connection, shortest length from input module to output module reflects also where we skip. But we can only distinguish the case of shortest length 1 and the other (because we have too many cases that coincidence on shortest length 2) which is not informative enough.

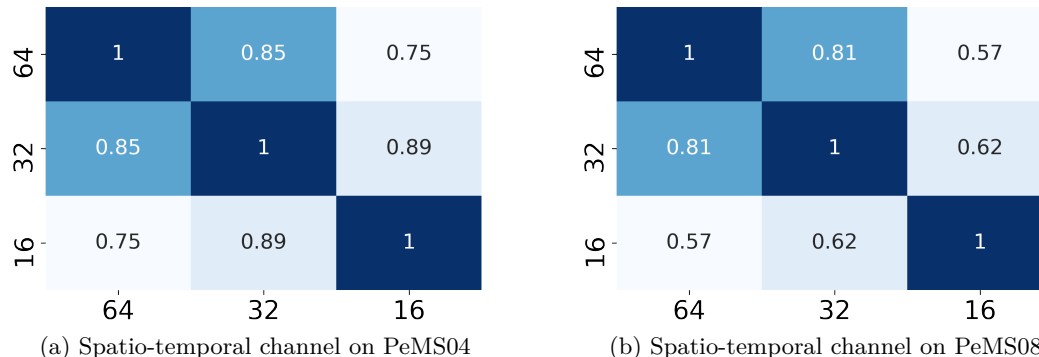

(a) Spatio-temporal channel on PeMS04       (b) Spatio-temporal channel on PeMS08

Figure 10: Performance correlation with different channels.

### D.5 More on the skip connection experiments

To better understand the impact of skip patterns, we fix the other parts of the framework. In this part, we study the case of 6 spatial/temporal modules thus $N$ is 8. As by calculation in Appendix C, we explore the skip space of $2^{21} = 2907152$ choices. For a certain dataset and a certain fixed the other configuration, we randomly sample among these two million choices and record their performance with the above mentioned measures.

## E Details on the searched model

### E.1 Configuration of searched model

The found best model on PeMS08 uses the following configuration. Learning rate 1e-3; Batch size 128; Optimizer AdamW; Weight decay 0; Gradient clip 5; Dropout 0.3; Curriculum learning 3 or 5; Temporal channels 64; Dilation 1; Kernet set 7,8; GCN channels 64; Graph convolution Mixhop GCN; Embedding dim 20; Node degree 40; Mixture coefficient 0.75.

### E.2 Evaluation metric

Following the literatue (Yu et al., 2017; Cui et al., 2020; Bai et al., 2020; Wu et al., 2020), we use past 12 steps to predict next 12 steps. Both metrisc MAE and RMSE are calculated as the average over future 12 steps. Each step is 5min so we are predicting one hour ahead using past hour. In Sec 5, we also give results on 3/6/12 steps (15/30/60 mins) to compare more thoroughly.

### E.3 More on the compared baselines

**MTGNN.** When running MTGNN, we notice that it applies a step-number-based curriculum learning. If we apply directly on other datasets of different size, it will not be able to train on all 12 prediction steps and perform poorly. As a result, we increase the epochs to 200 instead of 100 to have MTGNN trained on full 12 horizons. We leave other hyperparameters as default. Our executed results are close to the reproduced ones from AutoCTS (Wu et al., 2021).

**AutoSTG.** We find two works of AutoML applied to STGNN modeling, AutoCTS and AutoSTG. We compare only with AutoCTR because AutoSTG requires additional meta information from geographic database to construct their model which largely limits their applicability, as noted by AutoCTS (Wu et al., 2021).

### E.4 Parameter comparison

During our study, we also notice that the searched models are larger in terms of trainable parameters. In the literature, STGNN models are often smaller than 1 million(M) parameters while our searched model has parameters of 1.7M. To compare more fairly and understand better the models, we experiment further and show two things (1) more parameters do not always help (2) by compressing our searched model to same level of parameters with literature, we can still outperform them indicating our other choices of architecture indeed help. The results is summarized in Table 6.

Table 6: Prediction performance comparison with models of different scale (Dataset PeMS08).

| Model | #Param | MAE | RMSE |
|---|---|---|---|
| AGCRN | 0.8M | 15.95 | 25.22 |
| AGCRN (large) | 1.8M | 16.56 | 26.43 |
| MTGNN | 0.4M | 15.86 | 24.72 |
| MTGNN (large) | 1.9M | 16.02 | 25.30 |
| Our searched model (ch=16) | 0.4M | 15.13 | 24.13 |
| Our searched model (ch=32) | 0.8M | 15.04 | 24.03 |
| Our searched model (ch=64) | 1.7M | 14.75 | 23.82 |

### E.5 Evaluation on non-traffic dataset

To even further evaluate the generalization of our method, we use another non-traffic dataset "Electricity" (Wu et al., 2021) and measure on 3/6/12 steps as well as average over 12 steps.

Table 7: Performance comparison on Electricity.

| Model | Average | | 3 steps | | 6 steps | | 12 steps | |
|---|---|---|---|---|---|---|---|---|
| | MAE | RMSE | MAE | RMSE | MAE | RMSE | MAE | RMSE |
| STGCN | 423.1 | 3923.9 | 350.5 | 3706.5 | 414.3 | 3918.4 | 479.8 | 4111.2 |
| DCRNN | 400.5 | 3640.3 | 320.6 | 3356.5 | 380.8 | 3634.3 | 461.2 | 3919.6 |
| AGCRN | 297.1 | 2353.1 | 274.6 | 2066.6 | 304.3 | 2329.4 | 312.4 | 2475.4 |
| MTGNN | 261.0 | 2624.4 | 238.6 | 2331.3 | 267.9 | 2631.5 | 283.9 | 2867.7 |
| AutoCTS | 255.4 | 2101.6 | 230.9 | 1986.5 | 258.4 | 2115.9 | 277.1 | 2256.3 |
| **SimpleSTG (ours)** | **222.9** | **1926.9** | **214.8** | **1835.7** | **218.5** | **1843.7** | **238.0** | **2083.6** |
| Improvement(%) | 12.7 | 8.3 | 7 | 7.6 | 15.4 | 12.9 | 14.1 | 7.7 |

