# OpenReview forum: "Understanding and Simplifying Architecture Search in Spatio-Temporal Graph Neural Networks"
_TMLR — Accepted by TMLR_

### Review · Reviewer_J7Cd · 2022-11-20

**Summary Of Contributions:**

This paper studies automated architecture design for Spatio-Temporal Graph Neural Networks.
Concretely, the author identifies four factors of the search space design for this particular architecture, including hyperparameters, Spatio-temporal orders, skip connections, and spatial and temporal modules.
Then the author studies the optimal range of these factors respectively, thereby reducing their choices and hence the search space size.
The resulting search space is efficient enough that a simple random search produces strong results.
The proposed method can be viewed as a search space pruning technique that is generically applicable to NAS.
Empirically, the discovered architecture outperforms prior searched architectures on 6 out of 8 scenarios, but the results are stronger than it seems as the author considers transfer setting, whereas prior work tailors one architecture for every dataset.

**Audience:**

Yes

**Claims And Evidence:**

Yes

**Requested Changes:**

- "hyperparamters" typo in 4.1 para 1
- the distributional formulation of the NAS objective resembles DrNAS [1]. I encourage the author to mention it as well.

[1] Chen et al. DrNAS: Dirichlet Neural Architecture Search. ICLR 2021.

**Strengths And Weaknesses:**

Strength:

- The paper conducts a comprehensive study to identify the optimal range for each factor in the search space. The resulting reduced search space becomes very effective that a simple random search produces strong results.
- The study itself can be considered as a generic preprocessing step for NAS search space reduction, which by itself serves as a contribution to the community.
- Empirically, the discovered architecture demonstrates mostly superior performance compared with the prior NAS method, taking into the fact that the author considers the transfer setting, which is more practical but puts the proposed method at a potential disadvantage.

weakness:

- Disentangling factors during search space pruning does improve efficiency. However, doing it also discards conditioning between factors, if I understand it correctly. For example, the optimal weight decay distribution depends on the learning rate.
- I might've missed a point here about the computing ranking distributions for each factor. When studying one factor, how do you set the other factor? More concretely, take hyperparameters as an example; When plotting Figure 2.a, are other factors (e.g., skip connection) sampled at random?

---

### Review · Reviewer_D4kp · 2022-11-22

**Summary Of Contributions:**

This paper focuses on improving STGNN with Neural Architecture Search (NAS). It first provides several guidelines for designing STGNNs and then presents a NAS-based algorithm based on the proposed rules.

**Audience:**

Yes

**Claims And Evidence:**

No

**Requested Changes:**

Please refer to W2 and W3

**Strengths And Weaknesses:**

Strengths:
S1. This paper is well-written and easy to follow.

S2. This manuscript conducts extensive experiments and concludes with some meaningful rules.

S3. The new proposed algorithm achieves significant improvements beyond Non-NAS methods.

Weaknesses:
W1. The technical contribution of this article seems weak. From my experience in designing STGNNs, the reduced scope of hyperparameters is actually the best practice, and the same situation also occurs in the following topics. Therefore, it does not give me any extra information. This work is more like a technical report.

W2. The results are not very convincing. It is common that the random seed affects neural networks a lot. Compared with a worse seed, a better seed may achieve over 20% improvements. For each set of hyperparameters, it would be good if the experiments can be repeated with several different random seeds. Moreover, statistical significance should be provided as well.

W3. This article lacks an analysis of time consumption.

W4. As far as I know, there are many studies modeling the spatio-temporal dependencies in a joint manner, instead of factorizing them. It would be good to add this joint ST module into the search space.

Minor issues:

W5. The paper claims that we do not include parallel connections here because such designs do not perform well. However, it doesn't provide any evidence here.

---

### Review · Reviewer_dhML · 2022-12-03

**Summary Of Contributions:**

This paper highlights two issues in the current NAS literature for spatial-temporal (ST) problems: 1) Hyperparameters are not considered in the NAS framework. 2) The search space is too large. Instead, the authors proposed a disentangled Spatial-Temporal GNN framework that takes into account training hyperparameter search and introduced four additional search components: Spatial-Temporal module order, Skip connection, Spatial module, and Temporal module. Later, comprehensive studies and detailed analysis were conducted for each component. Lastly, they used these empirical findings to refine the hyperparameter search space and used random search on the reduced space to find the optimal network. Their results demonstrated that the proposed framework outperforms the existing baselines by hand-designed and automatically searched networks.

**Audience:**

Yes

**Claims And Evidence:**

Yes

**Requested Changes:**

This is an interesting work, however, I would like the authors to address some of my concerns in the weakness, such as 1, 2, 4, 6.

**Strengths And Weaknesses:**

Strength:
- The authors have addressed the issues raised by previous works and have taken training hyperparameters search into account for their NAS framework.
- Additionally, they proposed four key modules for their NAS framework and conducted comprehensive studies for each one, which provides some insight to understand the
 behavior of architectures.
- The authors utilized the above findings to minimize the hyperparameter search space and their experimental results demonstrate the superior performance compared to the most recent baseline approaches.

Weaknesses:
- At a high level, I am somewhat confused by the framework's design, particularly for Section 4. If we use a different dataset, do we need to repeat all experiments to find a reduced search space?
- In Section 2, the problem formulation is not well-explained and seems confusing to me. For example, what is the definition/formulation for L? Why is $D_{train}$ in the inner problem of bi-level formulation and $D_{val}$ in the outer problem?
- In Figure 2 and 5, the outer shape of each error bar is confusing to me. What does this irregular shape mean? Some explanations regarding the notations and interpretation need to be provided.
- The conclusion regarding transferability lacks sufficient support. The authors used two highway traffic flow datasets for experiments to validate the transferability. Nevertheless, these datasets are too similar (although with different regions, timestamps, and sensors).
- Table 3 lacks proper notation. The authors should mention or explain the bold and underscored numbers.
- Time comparison of different NAS approaches. AutoCTS seems to have a competitive performance; it would be interesting to compare the actual time cost for each approach to find the optimal network.

---

### Review · Reviewer_ZRsh · 2022-12-12

**Summary Of Contributions:**

The authors propose a Neural Architecture Search (NAS) method to design Spatio-Temporal Graph Neural Networks (STGNNs). They mainly deal with two issues. Firstly, they propose a fair and fast comparison of diverse architectures by integrating the examination of training hyperparameters into their NAS framework. Secondly, they solve the problem of large-scale search space by disentangling it into three disjoint angles. Their method is compared to several hand designed architectures and a searched architecture.

**Audience:**

Yes

**Broader Impact Concerns:**

No further concerns.

**Claims And Evidence:**

Yes

**Requested Changes:**

Please address problems in "Weaknesses" as mentioned above.

**Strengths And Weaknesses:**

Strengths:

- In the design of their framework, the authors consider and analyze much prior knowledge special for STGNNs, such as the spatio-temporal order.

- The efficiency of evaluation and fairness of comparison is crucial for NAS methods. Their integration of training hyperparameters into the search framework is necessary and meaningful.

Weaknesses:

- The authors introduce the concept of "cherry region", which seems interesting. However, it is rarely discussed in the paper (only mentioned twice). Although empirical evidence is provided in Figures 3 and 4, it would be better to also provide more in-depth insights into them. For example, why do those regions exist, does this phenomenon widely exist, and how can this phenomenon influence future studies?
- There is only one NAS method (i.e., AutoCTS) in the performance comparison in Table 3, which makes the experiment relatively weak. It would be better to include and discuss more NAS methods for GNNs, such as [1, 2].

Reference:

[1] Y. Gao, et al., "Graph Neural Architecture Search", In IJCAI, 2020.

[2] Y. Li, et al., "One-shot Graph Neural Architecture Search with Dynamic Search Space", In AAAI, 2021.

---

### Decision · Action_Editors · 2023-01-14

**Recommendation:** Accept as is

**Comment:**

This paper introduced neural architecture search for the design of  Spatio-Temporal Graph Neural Networks (STGNNs).  The authors highlighted the importance of hyperparameters in STGNNs and conducted comprehensive experiments for the evaluation. Another contribution is to address the large search space issue through a random search on the refined search space. All reviewers are happy with the authors' responses. But also some concerns were pointed out, e.g., the limited improvement over baseline AutoCTS. The authors noticed the literature on NAS for pure GNNs, but explained the difference between search spaces prevents a possible comparison. It is true that these existing NAS works are mostly about search strategies, while this paper focuses more on the search space. But it would be much more complete and convincing to include some relevant discussion or empirical study to suggest that a study on search space in this paper is really necessary and of significance, by comparing with those works on search strategies.

**Audience:**

This paper provides some interesting ideas for the graph convolution neural network and neural architecture search fields.

**Claims And Evidence:**

The overall paper was well written and the authors have well addressed the reviewers' comments.